# Blood multiomics reveal insights into population clusters with low prevalence of diabetes, dyslipidemia and hypertension

**Ming-Wei Su[1]☯, Chung-ke Chang[1]☯, Chien-Wei Lin[1], Shiu-Jie Ling[2], Chia-Ni Hsiung[1,3], Hou-Wei Chu[1], Pei-Ei Wu[1], Chen-Yang Shen[1,4]***

**1** Institute of Biomedical Sciences, Academia Sinica, Taipei, Taiwan, **2** Wego Private Bilingual Senior High School, Taipei, Taiwan, **3** Institute of Bioinformatics and Structural Biology, National Tsing Hua University, Hsinchu, Taiwan, **4** College of Public Health, China Medical University, Taichung, Taiwan

☯ These authors contributed equally to this work.
* bmcys@ibms.sinica.edu.tw

**Data Availability Statement:** The data that support the findings of this study were obtained from the Taiwan Biobank although restrictions apply to the availability of these data, which were used under

## Abstract

Diabetes, dyslipidemia and hypertension are important metabolic diseases that impose a great burden on many populations worldwide. However, certain population strata have reduced prevalence for all three diseases, but the underlying mechanisms are poorly understood. We sought to identify the phenotypic, genomic and metabolomic characteristics of the low-prevalence population to gain insights into possible innate non-susceptibility against metabolic diseases. We performed *k*-means cluster analysis of 16,792 subjects using anthropometric and clinical biochemistry data collected by the Taiwan Biobank. Nuclear magnetic resonance spectra-based metabolome analysis was carried out for 217 subjects with normal body mass index, good exercise habits and healthy lifestyles. We found that the gene *APOA5* was significantly associated with reduced prevalence of disease, and lesser associations included the genes *HIF1A*, *LIMA1*, *LPL*, *MLXIPL*, and *TRPC4*. Blood plasma of subjects belonging to the low disease prevalence cluster exhibited lowered levels of the GlycA inflammation marker, very low-density lipoprotein and low-density lipoprotein cholesterol, triglycerides, valine and leucine compared to controls. Literature mining revealed that these genes and metabolites are biochemically linked, with the linkage between lipoprotein metabolism and inflammation being particularly prominent. The combination of phenomic, genomic and metabolomic analysis may also be applied towards the study of metabolic disease prevalence in other populations.

## Introduction

Diabetes, dyslipidemia and hypertension are three common metabolic disorders affecting people middle-aged or older. According to World Health Organization estimates, diabetes was the direct cause of 1.6 million deaths in 2014, and high blood glucose accounted for another 2.2 million deaths in 2012 [1]. Dyslipidemia accounts for more than 4 million deaths per year [2]. Hypertension, or elevated blood pressure, is also estimated to cause more than 7.5 million

license for the current study and so are not publicly available. However, the data are available from the authors upon reasonable request or directly from the Taiwan Biobank (biobank@gate.sinica.edu.tw) pending permission from the Ministry of Health and Welfare, Taiwan.

**Funding:** This study was funded by the Institute of Biomedical Sciences, Academia Sinica. The funders had no role in the study design, data collection and analysis, decision to publish, or preparation of the manuscript.

**Competing interests:** The authors have declared that no competing interests exist.

deaths per year [3]. In addition to the high number of mortalities, diabetes, dyslipidemia and hypertension have relatively high prevalence rates in most populations, with global prevalence rates of 8.5% for diabetes [1], 38% for dyslipidemia [2], and 40% for hypertension [3]. Even more alarming is the fact that prevalence rates for all three diseases have increased steadily over time [4]. The three diseases already place an immense burden on public health systems worldwide, and the burden is projected to increase in the future.

Much effort has been made to elucidate the pathogenic mechanisms of the three diseases, which has resulted in a vast body of literature and advanced our understanding of their molecular underpinnings. However, these studies also underscore the heterogeneity and complexity of disease progression, where multiple mechanisms may alone or in conjunction lead to the same disease outcome. To complicate matters, lifestyle factors such as diet and exercise habits may also affect the disease status. In addition, diabetes, dyslipidemia and hypertension are interrelated [5]. There is also mounting evidence linking all three diseases at the biochemical and even genetic level [6], and pharmaceuticals targeted at one of the three diseases are often also effective against the other two [7, 8]. Similarly, subjects belonging to low prevalence population groups for the three diseases may share common phenotypes. Therefore, identification and investigation of these phenotypes and their underlying genetic and biochemical pathways may improve our understanding of the mechanisms involved in all or each of the three diseases.

Phenotypic cluster analysis has been widely adopted for subtyping individual diseases but has seldom been applied towards the identification of subjects belonging to low disease prevalence populations [9–11]. Because disease prevalence may be used to infer disease incidence, population clusters with reduced disease prevalence may potentially represent individuals with reduced disease susceptibility [12, 13]. We applied phenotypic cluster analysis on a subset of the Taiwan Biobank cohort and identified a cluster of subjects with low prevalence rates for all three diseases (diabetes, dyslipidemia and hypertension) based on self-reported disease status. We then took advantage of the integrated phenomic, genomic and metabolomic data available in the Taiwan Biobank to further infer that individuals belonging to the low-prevalence cluster generally had better lipid metabolism characteristics and less inflammation compared with controls. Several of the genes and metabolites identified in our study underscore the linkage between the two pathways and demonstrate that the mechanisms underlying these three diseases can be investigated via multiomics studies of population cohorts.

## Results

### Cohort characteristics

Blood sample genotype and clinical data of individual subjects were obtained from the Taiwan Biobank, and an overview is shown in Table 1. Individuals exhibiting trait values outside three times the interquartile range or four standard deviations were excluded from analysis. Of a total of 24,164 individuals, 16,792 were used for phenotypic clustering analysis. The mean values of most traits fell within the normal range, with the exception of low-density lipoprotein (LDL) cholesterol (121 mg/dL), which was higher than the level recommended by the National Cholesterol Education Program of the United States (< 100 mg/dL) [14].

### Phenotypic clustering reveals subjects with lower disease prevalence

We first conducted a PCA of study subjects using a set of anthropometric and biochemical phenotypes as variables. The variables are listed in Table 1. Positive disease status was assigned to respondents who reported that they have been diagnosed with a specific disease in the past. Fig 1A presents biplots of subjects with self-reported diabetes, dyslipidemia and/or

**Table 1. Overview of the studied traits.**

| Trait | Cluster 1 n = 4,405 | Cluster 2 n = 4,496 | Cluster 3 n = 4,401 | Cluster 4 n = 3,490 | Total n = 16,792 |
|---|---|---|---|---|---|
| Age (yr), mean (SD) | 48.8 (11.0) | 47.9 (11.2) | 48.3 (11.1) | 48.2 (10.9) | 48.3 (11.0) |
| Male, n (%) | 2251 (51.1) | 2248 (50.0) | 2121 (48.2) | 1786 (51.2) | 8406 (50.1) |
| BMI, mean (SD) | 24.4 (2.92) | 21.6 (2.54) | 23.3 (2.60) | 27.1 (3.16) | 23.9 (3.39) |
| Total cholesterol (mg/dL), mean (SD) | 185 (31.8) | 183 (30.7) | 199 (32.7) | 203 (33.8) | 192 (33.3) |
| HDL (mg/dL), mean (SD) | 48.7 (9.92) | 62.3 (12.4) | 55.4 (12.1) | 46.8 (9.54) | 53.7 (12.7) |
| LDL (mg/dL), mean (SD) | 118 (28.2) | 107 (26.7) | 127 (29.3) | 133 (30.2) | 121 (30.1) |
| Triglyceride (mg/dL), mean (SD) | 113 (56.1) | 64.6 (24.3) | 99.3 (45.7) | 152 (68.4) | 104 (58.4) |
| Fasting Glucose (mg/dL), mean (SD) | 92.2 (8.04) | 90.1 (7.56) | 91.6 (7.23) | 97.5 (10.7) | 92.6 (8.75) |
| HbA1c (%) | 5.61 (0.365) | 5.46 (0.334) | 5.50 (0.328) | 5.84 (0.450) | 5.59 (0.394) |
| Uric acid (mg/dL), mean (SD) | 5.44 (1.31) | 4.94 (1.19) | 5.77 (1.38) | 6.37 (1.44) | 5.59 (1.42) |
| eGFR (mL/min), mean (SD) | 106 (13.2) | 106 (12.5) | 99.2 (14.3) | 102 (14.0) | 103 (13.8) |
| Platelet (1000/uL), mean (SD) | 255 (56.9) | 222 (48.4) | 226 (47.9) | 253 (56.5) | 238 (54.5) |
| Systolic blood pressure (mmHg), mean (SD) | 117 (15.6) | 111 (15.3) | 115 (16.2) | 124 (16.5) | 116 (16.5) |
| Diastolic blood pressure (mmHg), mean (SD) | 72.3 (9.96) | 68.9 (9.68) | 72.1 (10.4) | 77.7 (10.6) | 72.4 (10.6) |
| Total bilirubin (mg/dL), mean (SD) | 0.56 (0.21) | 0.75 (0.25) | 0.74 (0.24) | 0.62 (0.23) | 0.66 (0.24) |
| Dyslipidemia, n (%) | 242 (5.5) | 123 (2.7) | 237 (5.4) | 339 (9.7) | 941 (5.6) |
| Hypertension, n (%) | 450 (10.2) | 195 (4.3) | 385 (8.7) | 596 (17.1) | 1626 (9.7) |
| Diabetes (%), n (%) | 113 (2.6) | 49 (1.1) | 42 (1.0) | 155 (4.4) | 359 (2.1) |

Cluster 1: Subjects with moderate prevalence of dyslipidemia and moderately high prevalence of hypertension and diabetes; Cluster 2: Relatively healthy subjects; Cluster 3: Subjects with moderate prevalence of dyslipidemia, hypertension and diabetes; Cluster 4: Subjects with high prevalence of dyslipidemia, hypertension and diabetes

hypertension as compared with the controls. Subjects with disease tended to cluster on the left side of the albumin (ALB) and estimated glomerular flowrate (eGFR) vectors. The biplot vectors representing the risk-factor traits associated with the three diseases (e.g. blood triglyceride concentration for dyslipidemia; fasting blood glucose for diabetes; systolic blood pressure for hypertension) point toward the same general direction. Only high-density lipoprotein cholesterol (HDL) content was inversely associated with the three diseases, which is consistent with its proposed role as a protective factor [15]. The co-directionality of risk and protective factors for diabetes, dyslipidemia and hypertension may reflect the shared underlying mechanisms of these three diseases [6]. Visual inspection revealed a region on the right side of the plot with a larger number of subjects free of diabetes, dyslipidemia and hypertension.

Using a *k*-means clustering approach, we found that our subjects fell into four distinct phenotypic clusters (Fig 1B). Table 1 presents a description of the phenotypic traits for each cluster. Clusters 1 and 3 had comparable rates of self-reported dyslipidemia, but Cluster 1 had a relatively higher prevalence of self-reported hypertension and diabetes. The main factors differentiating these two clusters are total serum bilirubin (T_BIL) and albumin (ALB), each of which has been associated with diabetes and hypertension as a protective factor in other populations [16, 17]. Our results confirm these previous observations in Taiwanese subjects of Han Chinese descent.

When compared with Fig 1A, we found that a large proportion of self-reported disease cases are located in Cluster 4. In addition to the known risk factors for diabetes, dyslipidemia and hypertension, subjects in Cluster 4 were also positively associated with the levels of γ-glutamyltransferase (γ GT), serum glutamic-pyruvic transaminase (SGPT, also called alanine aminotransferase, ALT) and uric acid (URATE) as well as white blood cell count (WBC). These

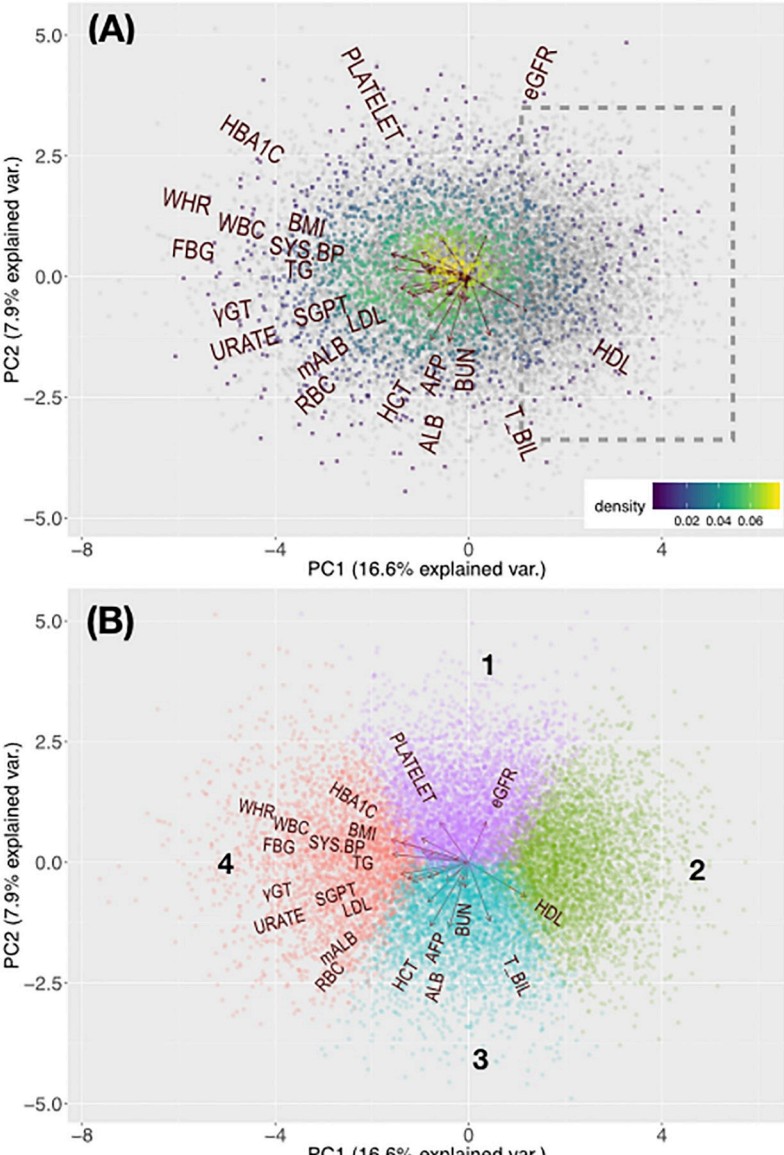

**Fig 1. Principal component analysis of study subjects.** (A) Biplot of study subjects based on anthropometric and clinical biochemistry data. Each dot represents an individual. Disease-free individuals are colored in grey and self-reported disease cases are denoted with gradient colors, which reflect dot density. The dot density was calculated by splitting the plot into a 500 × 500 grid of squares and counting the number of dots per each square. Warmer colors represent higher densities of individuals. Trait vectors are also shown. The dashed bounded box represents a region of individuals with low disease prevalence. Biplots for individual diseases are available in S4 Fig. (B) Clustering results mapped on the PCA biplot. Trait vectors and assigned cluster numbers are shown. eGFR: estimated glomerular filtration rate; PLATELET: platelet count; HBA1C: serum glycated hemoglobin concentration; WHR: waist-to-hip ratio; BMI: body mass index; WBC: white blood cell count; FBG: serum fasting blood glucose concentration; SYS.BP: systolic blood pressure; TG: serum triglyceride concentration; γGT: serum γ-glutamyltransferase concentration; SGPT: serum glutamic-pyruvic transaminase concentration; URATE: serum uric acid concentration; LDL: serum low-density lipoprotein cholesterol concentration; mALB: urine microalbumin concentration; RBC: red blood cell count; HCT: hematocrit; AFP: serum α-fetoprotein concentration; ALB: serum albumin concentration; BUN: blood urea nitrogen concentration; T_BIL: serum total bilirubin concentration; HDL: serum high-density lipoprotein cholesterol concentration.

traits have all been associated with cardiovascular disease, diabetes or obesity [18–21]. In agreement with our previous observation (Fig 1A), HDL was negatively associated with Cluster 4.

Cluster 2 is probably the most intriguing since it coincides with the segment of relatively healthy subjects identified by PCA (Fig 1A). Cluster 2 resides on the opposite side of Cluster 4 and is characterized by low levels of risk-factor traits for diabetes, dyslipidemia and hypertension, consistent with the low prevalence of these disorders within the cluster (Table 1).

## Common phenotypic traits define disease status in different clusters

There are large differences in the phenotypic traits defining the different clusters, but even the low-disease-prevalence Cluster 2 still contains subjects afflicted with diabetes, dyslipidemia or hypertension. To gain a better understanding of the phenotypic characteristics defining the disease states, we examined the average phenotypic values of diseased and control individuals in Cluster 2 and compared them to those of the high-disease-prevalence Cluster 4 (Table 2). Interestingly, although the average values for the traits differed widely between Clusters 2 and 4, individuals afflicted with disease shared a common phenotypic profile across these two clusters, i.e. their average values for risk-factor traits were higher than those of controls within the same cluster. The surprising exception was body fat rate: individuals with disease had reduced body fat rates when compared with controls within the same cluster. These results, combined with the observed higher blood triglyceride levels, suggest that impaired lipid metabolism is a trait that is shared by diseased subjects in Clusters 2 and 4. Our findings pertaining to risk-factor traits are consistent with common knowledge when only a single cluster is considered, but the large trait value discrepancies between clusters may affect the interpretation if clustering is not performed *a priori*. The interpretation of risk-factor trait values may thus need to be taken in the context of the phenotypic cluster under discussion.

## Genes associated with lower disease prevalence

We attempted to identify genes associated with Cluster 2 through a genome-wide association study (GWAS; Fig 2). The GWAS compared Cluster-2 samples with those of the other three clusters. Samples with call rates $< 0.95$ or heterozygosity rates larger than five standard deviations from the population mean were removed from the analysis, as were second relatives and non–East Asian outliers. A total of 16,710 samples remained after outliers were removed. For imputation, 4,597,401 SNPs (single-nucleotide polymorphisms) were used, and the imputation results were then filtered through the same quality control criteria used for outlier removal. Ultimately, 9,972,512 qualified SNPs were used for the final GWAS. Three SNPs were highly significant ($p < 5 \times 10^{-8}$): *rs651821*, *rs6494835* and *rs8035009* (Fig 2). Only *rs651821* is located in a known gene, which encodes apolipoprotein A5 (*APOA5*). Our results indicated that the minor allele frequency of the C allele of *rs651821* was significantly lower for Cluster 2 (0.2419) compared with the other clusters (Cluster 1 = 0.2843, Cluster 3 = 0.2715, Cluster 4 = 0.2877), suggesting that the C allele is associated with the disease state in our study population. We relaxed the significance threshold to $p < 10^{-5}$ and found five additional SNPs located in the following known genes: *LPL* (lipoprotein lipase), *MLXIPL* (MLX interacting protein-like), *HIF1A* (hypoxia inducible factor 1 subunit α), *TRPC4* (transient receptor potential cation channel subfamily C member 4) and *LIMA1* (LIM domain and actin-binding 1) (S1 Table). Many of these genes are associated with lipid and lipoprotein metabolism [22, 23]. The association of *APOA5* and other genes with Cluster 2 is consistent with the more normal lipoprotein profile of the cluster and suggests a direct relationship between the phenome and the genome.

**Table 2. Comparison of phenotypic values between Clusters 2 and 4.**

| | Cluster 2 | | | Cluster 4 | | |
|---|---|---|---|---|---|---|
| | **FALSE** | **TRUE** | **p** | **FALSE** | **TRUE** | **p** |
| n | 4172 | 324 | | 2661 | 829 | |
| SEX (mean (SD)) | 1.51 (0.50) | 1.31 (0.46) | <0.001 | 1.51 (0.50) | 1.40 (0.49) | <0.001 |
| AGE (mean (SD)) | 47.08 (11.00) | 58.21 (8.46) | <0.001 | 45.95 (10.31) | 55.59 (9.16) | <0.001 |
| BAI (mean (SD)) | 4.42 (0.34) | 4.43 (0.33) | 0.401 | 4.82 (0.41) | 4.79 (0.39) | 0.086 |
| BMI (mean (SD)) | 21.54 (2.54) | 22.37 (2.50) | <0.001 | 27.00 (3.16) | 27.24 (3.15) | 0.057 |
| BODY_FAT_RATE (mean (SD)) | 22.42 (6.14) | 21.54 (5.81) | 0.015 | 31.90 (6.65) | 30.70 (6.59) | <0.001 |
| WHR (mean (SD)) | 0.82 (0.06) | 0.86 (0.06) | <0.001 | 0.90 (0.06) | 0.93 (0.05) | <0.001 |
| T_CHO (mean (SD)) | 182.57 (30.34) | 188.96 (34.44) | <0.001 | 204.57 (33.68) | 199.26 (33.84) | <0.001 |
| HDL_C (mean (SD)) | 62.36 (12.41) | 60.91 (12.87) | 0.045 | 47.08 (9.60) | 45.83 (9.30) | 0.001 |
| LDL_C (mean (SD)) | 106.98 (26.35) | 113.43 (30.59) | <0.001 | 134.79 (30.01) | 127.76 (30.36) | <0.001 |
| TG (mean (SD)) | 64.31 (24.27) | 68.61 (24.44) | 0.002 | 148.48 (67.74) | 162.38 (69.49) | <0.001 |
| FASTING_GLUCOSE (mean (SD)) | 89.74 (6.93) | 95.32 (12.04) | <0.001 | 95.96 (9.28) | 102.28 (13.25) | <0.001 |
| HBA1C (mean (SD)) | 5.44 (0.32) | 5.64 (0.40) | <0.001 | 5.77 (0.39) | 6.06 (0.54) | <0.001 |
| ALBUMIN (mean (SD)) | 4.53 (0.32) | 4.52 (0.20) | 0.501 | 4.62 (0.35) | 4.62 (0.21) | 0.81 |
| BUN (mean (SD)) | 12.82 (3.29) | 14.47 (3.41) | <0.001 | 12.77 (3.13) | 14.22 (3.34) | <0.001 |
| CREATININE (mean (SD)) | 0.71 (0.18) | 0.78 (0.18) | <0.001 | 0.75 (0.20) | 0.81 (0.20) | <0.001 |
| URIC_ACID (mean (SD)) | 4.92 (1.19) | 5.26 (1.15) | <0.001 | 6.32 (1.44) | 6.55 (1.45) | <0.001 |
| eGFR (mean (SD)) | 106.53 (12.23) | 96.28 (11.98) | <0.001 | 104.30 (13.27) | 94.50 (13.52) | <0.001 |
| microALB (mean (SD)) | 8.70 (6.59) | 10.27 (9.14) | <0.001 | 12.22 (8.80) | 14.41 (10.67) | <0.001 |
| T_BILIRUBIN (mean (SD)) | 0.71 (0.24) | 0.77 (0.24) | <0.001 | 0.61 (0.23) | 0.64 (0.23) | <0.001 |
| SGOT (mean (SD)) | 22.13 (5.37) | 23.86 (5.22) | <0.001 | 24.98 (6.87) | 26.41 (6.69) | <0.001 |
| SGPT (mean (SD)) | 16.56 (7.01) | 18.13 (6.86) | <0.001 | 29.77 (14.87) | 30.61 (13.62) | 0.147 |
| AFP (mean (SD)) | 2.24 (1.75) | 2.39 (1.63) | 0.138 | 2.52 (1.73) | 2.77 (1.70) | <0.001 |
| GAMMA_GT (mean (SD)) | 14.52 (7.89) | 15.86 (7.53) | 0.003 | 30.40 (16.69) | 31.15 (15.69) | 0.256 |
| WBC (mean (SD)) | 5.25 (1.29) | 5.22 (1.25) | 0.723 | 6.72 (1.58) | 6.68 (1.50) | 0.458 |
| RBC (mean (SD)) | 4.62 (0.46) | 4.65 (0.40) | 0.391 | 5.00 (0.49) | 4.96 (0.45) | 0.081 |
| HB (mean (SD)) | 13.62 (1.52) | 13.93 (1.16) | <0.001 | 14.50 (1.60) | 14.64 (1.34) | 0.02 |
| HCT (mean (SD)) | 41.85 (5.71) | 42.29 (3.29) | 0.178 | 44.86 (7.07) | 44.92 (3.95) | 0.82 |
| PLATELET (mean (SD)) | 222.67 (48.23) | 209.61 (48.58) | <0.001 | 257.14 (56.99) | 241.27 (53.29) | <0.001 |
| SYSTOLIC (mean (SD)) | 110.19 (14.52) | 125.41 (17.81) | <0.001 | 121.74 (15.63) | 132.59 (16.47) | <0.001 |
| DIASTOLIC (mean (SD)) | 68.43 (9.48) | 75.00 (10.13) | <0.001 | 76.79 (10.55) | 80.44 (10.28) | <0.001 |
| MAP (mean (SD)) | 127.97 (16.55) | 141.80 (18.21) | <0.001 | 142.96 (18.25) | 151.45 (17.51) | <0.001 |
| PP (mean (SD)) | 41.77 (9.81) | 50.42 (12.79) | <0.001 | 44.95 (10.58) | 52.15 (13.33) | <0.001 |

Abbreviations are the same as those used in Fig 1. TRUE represents subjects afflicted with diabetes, dyslipidemia or hypertension.

We further attempted to identify genes associated with the disease state within either Cluster 2 or Cluster 4 (S1 Fig). Although none of the SNPs in either GWAS reached statistical significance ($p < 5 \times 10^{-8}$) probably due to the rather small number of samples in each cluster, the Manhattan plots for the two clusters differed markedly, suggesting that the phenotypic clustering approach may also yield data concerning differences in disease-related genetic factors.

### Blood plasma metabolome characteristics of Cluster 2

To understand the possible biochemical mechanisms involved in the subjects of Cluster 2, we examined the blood plasma metabolome profiles of 144 Cluster 2 subjects and 73 control

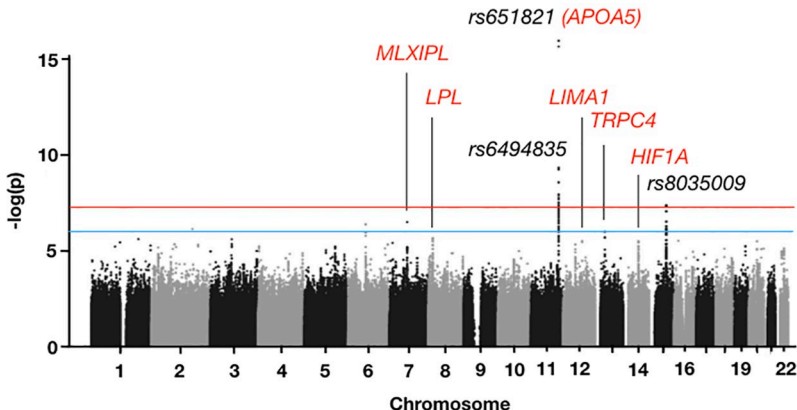

**Fig 2. Manhattan plot representing the SNPs associated with the Cluster 2 phenotype.** The three highly significant SNPs are shown in black. SNPs within known genes are represented by their gene names, colored in red.

subjects (i.e., those not associated with Cluster 2) using NMR spectroscopy (Fig 3). The stringent subject selection criteria reduced the number of confounders and allowed the straightforward analysis of the metabolomic results even with a limited sample size. The full spectrum is shown in S2 Fig. We first examined the far methyl region of the spectra (Fig 3A), which contains signals from lipoproteins and triglycerides [24]. Compared with the control subjects, Cluster 2 subjects had lower concentrations of very-low-density/low-density lipoprotein (VLDL/LDL) cholesterol and triglycerides, consistent with our phenotypic observations. We also observed a decrease in the concentrations of the branched-chain amino acid leucine, which has resonances located in this same region [25]. Moving the observation window

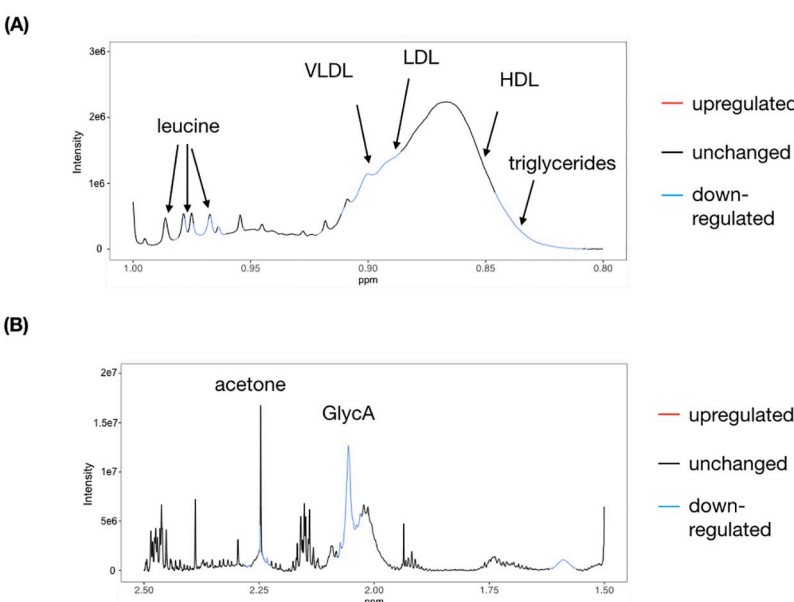

**Fig 3. NMR spectral regions associated with Cluster 2.** (A) Far-methyl region including signals from lipoproteins and branched-chain amino acids. (B) Downfield region from (A) which includes the GlycA inflammation marker and acetone.

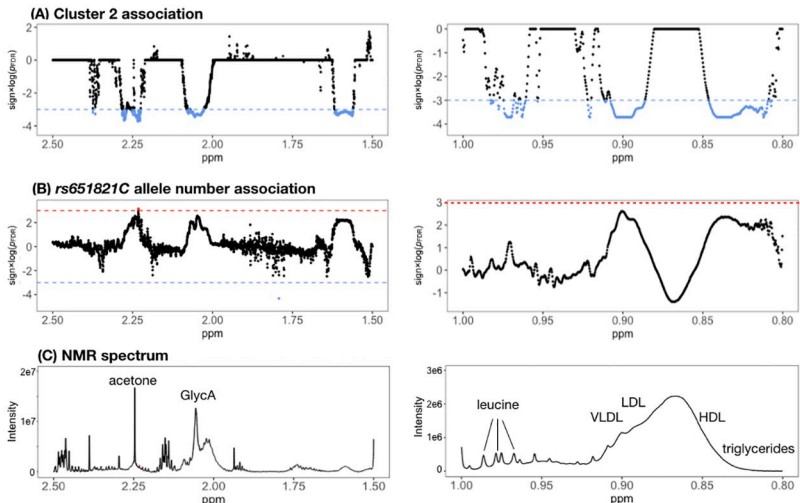

**Fig 4. FDR plot of the NMR metabolome association with Cluster 2 (A) and number of C alleles of *rs651821* (B).**
Red and blue horizontal lines represent the threshold for statistical significance in terms of up- or down-regulation of
metabolite levels, respectively. For reference, a representative NMR spectrum (including metabolite assignments) is
shown in (C).

downfield, we observed a reduction of the GlycA signal at ~2.04 ppm (Fig 3B), which is a com-
posite marker of systemic inflammation [26].

Because *rs651821* of *APOA5* appeared to be highly associated with lower disease prevalence,
we also examined its association with the metabolomic profile. We calculated the association
between the number of C alleles at *rs651821* and the NMR spectra. Although the associations
did not reach statistical significance, the false discovery rate (FDR) plot was highly correlated
with the one between Cluster 2 and NMR spectra (Fig 4). The FDR plot is similar to the Man-
hattan plot commonly used in genomics. Based on the FDR plots, lipoprotein and triglyceride
signals appeared to increase with the number of C alleles. We also noted concurrent changes
in the signals of leucine and GlycA. These results suggested that *APOA5* may directly or indi-
rectly affect inflammatory pathways in addition to its canonical role in lipid metabolism.

## Discussion

In this study, we used a combination of phenomics, genomics and metabolomics to character-
ize human subjects belonging to a population cluster having a low prevalence of diabetes, dys-
lipidemia and hypertension. The approach allowed us to identify probable links between the
genes, metabolites and phenotypes involved in the three metabolic diseases. Our findings
show that individuals belonging to the low-disease-prevalence cluster share a common genetic
background, which may translate to better blood plasma lipid profiles and reduced levels of
inflammation-inducing metabolites.

Of the genes we identified as being associated with low disease prevalence, i.e., *APOA5*,
*LPL*, *MLXIPL*, *LIMA1*, *HIF1A* and *TRPC4*, most are directly involved in lipid metabolism.
ApoA5 enhances the lipase activity of Lpl and may affect the kinetics of VLDL production
[23]. Indeed, our metabolomic results revealed reduced levels of both VLDL and triglycerides
in the individuals belonging to the low-prevalence cluster, and this may reflect differences in
*APOA5* genotype compared with controls. Because the metabolomic differences were evident
even after controlling for body mass index, smoking and drinking, and exercise habits, it is
very likely that the genetic component plays a definitive role in maintaining a healthy lipid

profile in individuals. Enhanced Lpl activity reportedly leads to better clearance of circulating triglycerides, which may translate to a lower risks of developing diabetes, dyslipidemia and/or hypertension [23]. *MLXIPL* encodes a transcription factor that activates glucose-dependent conversion of excess nutrients into triglycerides instead of glycogen [22]. *LIMA1* may modulate plasma low-density lipoprotein cholesterol level by regulating intestinal cholesterol absorption [27]. *HIF1A* is a master transcriptional regulator of genes encoding factors that govern lipid metabolism and inflammation, both of which are central mechanisms underlying the progression of metabolic disease [28, 29]. Hif1a enhances expression of the VLDL receptor, causing lipid accumulation in cells. It also assists in the differentiation of CD4$^+$ T cells into Th17, which are pro-inflammatory, instead of regulatory T cells, which are anti-inflammatory [30]. The remaining candidate gene, *TRPC4*, may play a role in diabetes, but the exact mechanism remains obscure [31].

Our metabolomic results corroborate our genetic findings. Fatty acids derived from triglycerides and VLDL can induce inflammation. We found that GlycA signal intensity appears to be associated with the *APOA5* genotype, suggesting that the inflammatory response may be linked to lipid metabolism. Reduced levels of triglycerides and VLDL could be detected easily in the NMR spectra of the low-disease-prevalence group and may explain the reduced inflammation status observed for those individuals.

In terms of small molecules, the branched chain amino acid leucine is particularly intriguing: free leucine in the circulation promotes glucose uptake by hepatic cells via myostatin, which in turn inhibits glycogenesis and promotes the synthesis of triglycerides and VLDL [32]. Furthermore, leucine acts as a mTORC1 activator, and in hepatic cells mTORC1 inhibits the CREBH-ApoA5 axis, which leads to blunted ApoA5 production [33]. Thus, a low concentration of leucine in the blood may have a dual effect in the liver—it reduces the production of VLDL/triglycerides yet enhances ApoA5 synthesis and assists in the clearance of VLDL/triglycerides from the circulation. Leucine also activates mTORC1 in the immune system, which in turn regulates Hif1a levels in T cells and monocytes, and Hif1a is indispensable for T cell activation [34]. It appears that the leucine-mTORC1-Hif1a pathway is a key link between lipid metabolism and inflammation and thus may be a prime target for drug development. For humans, leucine is an essential amino acid that is acquired mainly from ingested food; moreover, because leucine is a hydrophobic amino acid, it is most abundant in relatively high-fat foods. We hypothesize that individuals belonging to the low prevalence cluster for diabetes, dyslipidemia and hypertension generally do not experience the "double jeopardy" of elevated postprandial levels of circulating fatty acids and leucine, which may synergistically activate different inflammation pathways and thereby increase disease susceptibility. As such, individuals prone to these diseases may minimize their risk by adhering to a low-fat diet. Our results provide a framework for the design of further studies to test these possibilities.

Although phenotypic clustering has gained popularity as a tool for disease subtyping, few attempts have been made to utilize this approach with a population containing multiple, distinct diseases. Because the biochemical correlation amongst diabetes, dyslipidemia and hypertension is well established, our clustering results may not be surprising; however, our study suggests the intriguing possibility that biochemical relationships between different diseases may be inferred through population-based phenotypic cluster analysis. The ability to link these biochemical relationships to individual genes through GWAS provides another layer of information, especially in terms of possible pleiotropic effects for genes whose functions are known [35]. The gene-phenotype relationship can be further validated through the metabolome, which provides a bridge between the genetic and phenotypic information [36]. The metabolome may also provide additional clues to the biochemistry underlying a disease, i.e., if metabolite levels initially considered to be unrelated to the disease are, in fact, found to be

affected. Genes whose functions are related to these 'secondary' metabolites—especially those just below the threshold of statistical significance—may then be further scrutinized. The combination of phenotypic clustering, metabolome-wide association studies and GWAS provides a powerful approach for investigating diseases which are linked at the biochemical level. From a clinical viewpoint, this hybrid approach also may provide clues for possible treatment options. For example, healthy subjects belonging to Cluster 1 who later develop diabetes may have a prominent oxidative stress component as alluded to by the inverse association with serum albumin and bilirubin, both of which are well-established circulatory antioxidants [16, 17]. These individuals may benefit the most from antioxidant intervention, whereas similar subjects belonging to other clusters may be less responsive to the same treatment. Interestingly, the genetic profiles associated with disease may differ among clusters (S1 Fig), which indicates that these clusters may reflect slightly different disease mechanisms. This raises the possibility of exploiting these differences for the development of precision therapeutics for phenotypic clusters.

Our current study has several limitations. First, the use of self-reported disease status may severely underestimate the prevalence of hypertension, diabetes, and dyslipidemia in the population and bias our results. However, the information on disease status was used only to define the cluster characteristics and increasing the prevalence numbers should not affect our conclusions. Second, the number of subjects with available metabolomics data was relatively small, so our analysis could reveal only large effects, and more subtle metabolic changes may have been missed. Finally, our study was of an exploratory nature; therefore, elucidating any causal relationships between the numerous genes, metabolites and phenotypes we identified will require a large number of follow-up studies, probably utilizing animal models rather than human subjects.

## Materials and methods

A detailed Materials and Methods section is available in the S1 File. All experimental procedures and protocols were approved by the Institutional Review Board on Biomedical Science Research of Academia Sinica, Taiwan, and by the Ethics and Governance Council of the Taiwan Biobank. All research was performed in accordance with relevant guidelines. Written informed consent was obtained from all participants of the Taiwan Biobank for broad scientific use of the data. All data were anonymized prior to access.

We obtained blood sample genotype data and clinical information of 24,164 subjects from the Taiwan Biobank. Among these subjects, 400 also had nuclear magnetic resonance (NMR)-based blood metabolome data available. Disease status was obtained by asking respondents whether they have ever been diagnosed with diabetes, dyslipidemia and/or hypertension in the past.

### Phenotypic cluster analysis

The Taiwan Biobank collects anthropometric and clinical information on participants through standardized interview questionnaires and clinical biochemistry conducted at certified laboratories [37]. Quantitative traits used in this study included age, height, weight and clinical biochemistry data from assays typically conducted during routine health examinations. Details of the data processing steps, including missing value imputation, stratification by age and gender, and normalization of the values are described in the Online Methods. To minimize undue computational complexity during cluster analysis, we removed redundant, i.e., highly correlated, traits. A PCA with all trait values as the input was performed to generate a biplot and to calculate the principal component loadings of each trait. We then calculated the Pearson's

correlation coefficient between each pair of traits (S3 Fig). For each trait pair with a correlation coefficient of >0.6, we removed the trait with the lesser principal component loadings.

The remaining traits served as input for clustering using the *k*-means algorithm. The optimal number of clusters was determined using the average silhouette approach. To avoid cluster instability, the clustering algorithm was repeated 1,000 times with different random cluster centers, and the cluster similarity across repeats was assessed through the mean Jaccard index. The clustering results were visualized by overlaying cluster samples onto the PCA biplot.

## GWAS

The genotype data stored in the Taiwan Biobank were acquired on a previously described customized array using the Affymetrix Axiom platform [37]. The following types of subjects were excluded from analysis: those with call rates < 0.95, with heterozygosity rates of more than five standard deviations from the population mean, from closely related individuals, and individuals who were not of East Asian descent. We also excluded SNPs that had call rates < 0.95, minor allele frequencies < 5%, or Hardy–Weinberg equilibrium *p* values < $10^{-5}$. We prephased the genotypes with SHAPEIT v2.r790 (https://mathgen.stats.ox.ac.uk/genetics_software/shapeit/shapeit.html) and imputed dosages with IMPUTE2 v2.3.1 (http://mathgen.stats.ox.ac.uk/impute/impute_v2.html) using the 1000 Genomes Project Phase 3 (ftp://ftp.1000genomes.ebi.ac.uk/vol1/ftp/release/20130502/) East Asian haplotypes as reference. The imputation results were subjected to another round of quality control. Imputed SNPs with low imputation quality ($R_{sq} < 0.8$), high missing rates (> 5%), significant deviation from Hardy-Weinberg equilibrium ($p < 10^{-5}$), or low minor allele frequencies (< 0.1%) were excluded from the subsequent association analysis. Type 1 errors arising from population structure were corrected by applying PCA to the SNPs and using eight principal components as covariates in the regression model. The inflation factors were generally close to unity (0.95–1.01) after this operation. Genome-wide associations were determined under a logistic regression model and assuming additive allelic effects using PLINK v1.9 (https://www.cog-genomics.org/plink2/) with age, gender, BMI and the principal components as covariates. Genome-wide significance thresholds were initially set at $p = 5.0 \times 10^{-8}$ after Bonferroni correction. The threshold was later relaxed to $p = 5 \times 10^{-6}$ to identify a larger number of candidate genes.

## NMR-based metabolome profiling

Raw NMR free induction decay data acquired using the `cpmgpr1d` pulse sequence (Bruker, Germany) of 217 blood plasma specimens were obtained from the Taiwan Biobank. Specimens were chosen from fasting participants with the following criteria: (1) Body-mass index between 18 and 24, (2) Non-smokers not addicted to alcohol and (3) at least three exercise sessions per week lasting ≥30 minutes per session. This stringent selection process was implemented to avoid confounding the data. Details on NMR sample preparation are described in the Online Methods. A line broadening of 0.3 Hz was applied to each free induction decay data prior to applying the Fourier transform in Topspin v3.5pl7 (Bruker, Germany). The spectral phase was manually corrected within the program. The water region between 4–5 ppm was then removed from all spectra. Baseline correction and resonance signal alignment were carried out in the R statistical environment v3.5 (The R Foundation for Statistical Computing) with airPLS (https://github.com/zmzhang/airPLS) and SPEAQ v2.0 (https://cran.r-project.org/web/packages/speaq/index.html). The association of NMR signals with the low disease prevalence cluster (Cluster 2) was carried out with MWASTools (https://www.bioconductor.org/packages/3.7/bioc/html/MWASTools.html) using logistic regression with age and gender as confounders. Resonance signals showing statistically significant association, i.e., FDR < $10^{-3}$

after Benjamini-Yekutieli correction, were assigned by referencing previous publications and the Human Metabolome Database [24, 25, 38]. The linear association between the NMR signals and the number of C alleles of *rs651821*, which is correlated with disease (see Results), was analyzed using the same approach, but without correction for FDR.

## Supporting information

**S1 Fig. Manhattan plot of Cluster 2 and Cluster 4 comparing the GWAS results of subjects with disease to controls within the same cluster.** Red dots represent SNPs with $p < 10^{-6}$. (TIF)

**S2 Fig. Representative 1D $^1$H CPMG-PRESAT spectrum of blood plasma covering the full spectral width.** The water region (4–5 ppm) has been removed for clarity. (TIF)

**S3 Fig. Trait correlation heat map.** Blue and red colors indicate positive and negative correlations, respectively. Numbers denote the Pearson's correlation between the traits. (TIF)

**S4 Fig. Biplots of individual diseases and all diseases aggregated together.** (TIF)

**S1 Table. SNPs associated with low disease prevalence.** (DOCX)

**S1 File. Supplementary material and methods.** (DOCX)

## Acknowledgments

We acknowledge the Taiwan Biobank for excellent sample collection, storage and documentation. All NMR spectra were acquired at the High-Field NMR Center of Academia Sinica, Taiwan. This work was supported by the Institute of Biomedical Sciences, Academia Sinica, Taiwan.

## Author Contributions

**Conceptualization:** Ming-Wei Su, Chen-Yang Shen.

**Data curation:** Ming-Wei Su, Chung-ke Chang, Chien-Wei Lin, Chia-Ni Hsiung.

**Formal analysis:** Ming-Wei Su, Chung-ke Chang, Chien-Wei Lin, Shiu-Jie Ling.

**Funding acquisition:** Chen-Yang Shen.

**Investigation:** Ming-Wei Su, Chung-ke Chang, Shiu-Jie Ling.

**Methodology:** Ming-Wei Su, Chung-ke Chang, Chien-Wei Lin.

**Project administration:** Hou-Wei Chu, Pei-Ei Wu.

**Resources:** Chen-Yang Shen.

**Supervision:** Chia-Ni Hsiung, Chen-Yang Shen.

**Writing – original draft:** Ming-Wei Su, Chung-ke Chang, Chen-Yang Shen.

**Writing – review & editing:** Ming-Wei Su, Chung-ke Chang, Chen-Yang Shen.

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
