## [Decision Letter · Decision Letter 0]

19 Dec 2019

PONE-D-19-27166

Blood Multiomics Reveal Insights into Population Cluster with Low Prevalence of Diabetes, Dyslipidemia and Hypertension

PLOS ONE

Dear Dr. Chang,

Thank you for submitting your manuscript to PLOS ONE. After careful consideration, we feel that it has merit but does not fully meet PLOS ONE’s publication criteria as it currently stands. Therefore, we invite you to submit a revised version of the manuscript that addresses the points raised during the review process.

We would appreciate receiving your revised manuscript by Feb 02 2020 11:59PM. To enhance the reproducibility of your results, we recommend that if applicable you deposit your laboratory protocols in protocols.io, where a protocol can be assigned its own identifier (DOI) such that it can be cited independently in the future. For instructions see: http://journals.plos.org/plosone/s/submission-guidelines#loc-laboratory-protocols

We look forward to receiving your revised manuscript.

Kind regards,

Jumana Yousuf Al-Aama, MD, SBP, MRCP, FCCMG

Academic Editor

PLOS ONE

Journal Requirements:

Please ensure that your manuscript meets PLOS ONE's style requirements, including those for file naming. The PLOS ONE style templates can be found at http://www.plosone.org/attachments/PLOSOne_formatting_sample_main_body.pdf and http://www.plosone.org/attachments/PLOSOne_formatting_sample_title_authors_affiliations.pdfIn ethics statement in the manuscript and in the online submission form, please provide additional information about the patient records used in your retrospective study. Specifically, please ensure that you have discussed whether all data were fully anonymized before you accessed them and/or whether the IRB or ethics committee waived the requirement for informed consent. If patients provided informed written consent to have data from their medical records used in research, please include this information. We note that you have indicated that data from this study are available upon request. PLOS only allows data to be available upon request if there are legal or ethical restrictions on sharing data publicly. For more information on unacceptable data access restrictions, please see http://journals.plos.org/plosone/s/data-availability#loc-unacceptable-data-access-restrictions.In your revised cover letter, please address the following prompts:a) If there are ethical or legal restrictions on sharing a de-identified data set, please explain them in detail (e.g., data contain potentially sensitive information, data are owned by a third-party organization, etc.) and who has imposed them (e.g., an ethics committee). Please also provide contact information for a data access committee, ethics committee, or other institutional body to which data requests may be sent.b) If there are no restrictions, please upload the minimal anonymized data set necessary to replicate your study findings as either Supporting Information files or to a stable, public repository and provide us with the relevant URLs, DOIs, or accession numbers. For a list of acceptable repositories, please see http://journals.plos.org/plosone/s/data-availability#loc-recommended-repositories.We will update your Data Availability statement on your behalf to reflect the information you provide. Please include a copy of Table 2 which you refer to in your text on page 11. Please include captions for your Supporting Information files at the end of your manuscript, and update any in-text citations to match accordingly. Please see our Supporting Information guidelines for more information: http://journals.plos.org/plosone/s/supporting-information.

Reviewers' comments:

Reviewer's Responses to Questions

**Comments to the Author**

1. Is the manuscript technically sound, and do the data support the conclusions?

Reviewer #1: Yes

Reviewer #2: Yes

2. Has the statistical analysis been performed appropriately and rigorously? 

Reviewer #1: Yes

Reviewer #2: Yes

3. Have the authors made all data underlying the findings in their manuscript fully available?

Reviewer #1: Yes

Reviewer #2: Yes

4. Is the manuscript presented in an intelligible fashion and written in standard English?

Reviewer #1: Yes

Reviewer #2: Yes

5. Review Comments to the Author

Reviewer #1: This study aims to investigate the use of blood multiomics to reveal insights into population cluster with low prevalence of diabetes, dyslipidemia and hypertension. This is a very interesting work with well-organized format. I would recommend its acceptance after some minor corrections in English language.

Reviewer #2: Thank you for letting me review this interesting paper.

In current work, Su et al. used a large-scaled database from Taiwan Biobank to identify the phenotypic, genomic, and metabolomic characteristics of the low prevalence population to gain insights into possible innate non-susceptibility against metabolic diseases. In general, the paper is pretty well-written and the approach is comprehensive. The final metabolomic findings in Figure 4 also functionally and successfully validated the genomic findings. The authors concluded that the low disease prevalence cluster share a common genetic background, which may translate to better blood plasma lipid profiles and reduced levels of inflammation-inducing metabolites. They also provided clues into possible treatment options, as mentioned between line287-294.

Since right now in scientific field, multi-omic dataset context for certain disease is just emerging. This paper indicated a very good approach/example for future multi-omic direction and worth published in certain field, even though some limitations have already been discussed in the final paragraph of DISCUSSION.

Here I only raise some minor points for authors to revise in the future version:

1. line 146, "Table 2" is not attached.

2. line 168, "Table 1" is wrong and must be revised.

3. I strongly suggested a native English writer to edit the whole manuscript.

6. PLOS authors have the option to publish the peer review history of their article (what does this mean?). If published, this will include your full peer review and any attached files.

Reviewer #1: No

Reviewer #2: No

---

## [Author Response · Author response to Decision Letter 0]

20 Jan 2020

Journal format requirements:

Answer: We have made the appropriate changes according to the style templates.

2. In ethics statement in the manuscript and in the online submission form, please provide additional information about the patient records used in your retrospective study. Specifically, please ensure that you have discussed whether all data were fully anonymized before you accessed them and/or whether the IRB or ethics committee waived the requirement for informed consent. If patients provided informed written consent to have data from their medical records used in research, please include this information.

Answer: We did not utilize patient records from any medical institution, but rather relied on our subjects’ self-reported disease status, which is a standard multiple-choice question included in the Taiwan Biobank questionnaire. All the data were fully anonymized before access, and written informed consent from each and every Taiwan Biobank subject was given for the broad scientific use of the data. We have added the relevant parts to the first paragraph of the Materials and Methods section.

3. We note that you have indicated that data from this study are available upon request. PLOS only allows data to be available upon request if there are legal or ethical restrictions on sharing data publicly. For more information on unacceptable data access restrictions, please see http://journals.plos.org/plosone/s/data-availability#loc-unacceptable-data-access-restrictions. In your revised cover letter, please address the following prompts:

Answer: Data from the Taiwan Biobank is subject to the “Biobank Examination Guidelines for the International Transmission of Biobank Data and the Export of Biospecimen Derivatives” (temporary translation) issued by the Ministry of Health and Welfare of Taiwan (http://www.rootlaw.com.tw/LawArticle.aspx?LawID=A040170031038200-1060912 in Chinese). As such, access to biobank-derived data regardless of anonymization status, such as the ones used to conduct our study, by foreign entities requires explicit examination and permission by the Ministry. This includes the deposition of data to servers whose physical locations are outside of Taiwan. In order to be granted access, the party requesting the data may contact the authors or the Taiwan Biobank directly (biobank@gate.sinica.edu.tw), which may then submit an international data transfer application for review by the Ministry.

4. Please include a copy of Table 2 which you refer to in your text on page 11.

Answer: We apologize for the oversight. Table 2 has been included in the manuscript in the subsection titled “Common phenotypic traits define disease status in different clusters”.

Answer: The captions have been added and the in-text citations updated according to the guidelines.

Response to Reviewer #1 comments:

 This study aims to investigate the use of blood multiomics to reveal insights into population cluster with low prevalence of diabetes, dyslipidemia and hypertension. This is a very interesting work with well-organized format. I would recommend its acceptance after some minor corrections in English language.

 Answer: We have sent the manuscript to a professional English editor for English editing. The revised manuscript has incorporated all the recommended language changes.

Response to Reviewer #2 comments:

1. line 146, "Table 2" is not attached.

Answer: Attached in the revision. Again, we apologize for the oversight.

2. line 168, "Table 1" is wrong and must be revised.

Answer: We were referring to Fig 2 instead. Apologies for the oversight. We have also amended Fig 2 such that the connection between rs651821 and the APOA5 gene are better marked for the reader.

3. I strongly suggested a native English writer to edit the whole manuscript.

Answer: We have sent the manuscript to a professional English editor for English editing. The revised manuscript has incorporated all the recommended changes in language.

---

## [Decision Letter · Decision Letter 1]

19 Feb 2020

Blood multiomics reveal insights into population clusters with low prevalence of diabetes, dyslipidemia and hypertension

PONE-D-19-27166R1

Dear Dr. Chang,

We are pleased to inform you that your manuscript has been judged scientifically suitable for publication and will be formally accepted for publication once it complies with all outstanding technical requirements.

With kind regards,

Jumana Yousuf Al-Aama, MD, SBP, MRCP, FCCMG

Academic Editor

PLOS ONE

Additional Editor Comments (optional):

Reviewers' comments:

Reviewer's Responses to Questions

**Comments to the Author**

1. If the authors have adequately addressed your comments raised in a previous round of review and you feel that this manuscript is now acceptable for publication, you may indicate that here to bypass the “Comments to the Author” section, enter your conflict of interest statement in the “Confidential to Editor” section, and submit your "Accept" recommendation.

Reviewer #1: All comments have been addressed

2. Is the manuscript technically sound, and do the data support the conclusions?

Reviewer #1: Yes

3. Has the statistical analysis been performed appropriately and rigorously? 

Reviewer #1: Yes

4. Have the authors made all data underlying the findings in their manuscript fully available?

Reviewer #1: Yes

5. Is the manuscript presented in an intelligible fashion and written in standard English?

Reviewer #1: Yes

6. Review Comments to the Author

Reviewer #1: The author basically answered my all comments. The English language has been improved. Here, no more comments are provided.

7. PLOS authors have the option to publish the peer review history of their article (what does this mean?). If published, this will include your full peer review and any attached files.

Reviewer #1: No

---

## [Editor Report · Acceptance letter]

25 Feb 2020

PONE-D-19-27166R1 

Blood multiomics reveal insights into population clusters with low prevalence of diabetes, dyslipidemia and hypertension 

Dear Dr. Chang:

I am pleased to inform you that your manuscript has been deemed suitable for publication in PLOS ONE. Congratulations! Your manuscript is now with our production department. 

With kind regards,

on behalf of

Dr Jumana Yousuf Al-Aama 

Academic Editor

PLOS ONE